# Tracking Lake and Reservoir Changes in the Nenjiang Watershed, Northeast China: Patterns, Trends, and Drivers

**Baojia Du [1,2], Zongming Wang [1,3], Dehua Mao [1,*] 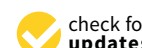, Huiying Li [4] and Hengxing Xiang [1,2]**

[1] Key Laboratory of Wetland Ecology and Environment, Northeast Institute of Geography and Agroecology, Chinese Academy of Sciences, Changchun 130102, China; dubaojia@neigae.ac.cn (B.D.); zongmingwang@neigae.ac.cn (Z.W.); xianghengxing@neigae.ac.cn (H.X.)
[2] University of Chinese Academy of Sciences, Beijing 100049, China
[3] National Earth System Science Data Center, Beijing 100101, China
[4] School of Management Engineering, Qingdao University of Technology, Qingdao 266033, China; lihy@qut.edu.cn
[*] Correspondence: maodehua@neigae.ac.cn; Tel.: +86-431-8554-2254; Fax: +86-431-8854-2298

**Abstract:** In terms of evident climate change and human activities, investigating changes in lakes and reservoirs is critical for sustainable protection of water resources and ecosystem management over the Nenjiang watershed (NJW), an eco-sensitive semi-arid region and the third-largest inland waterbody cluster in China. In this study, we established a multi-temporal dataset documenting lake and reservoir (area ≥ 1 km$^2$) changes in this region using an object-oriented image classification method and Landsat series images from 1980 to 2015. Using the structural equation model (SEM), we analyzed the diverse impacts of climatic and anthropogenic variables on lake changes. Results indicated that lakes experienced significant changes with fluctuations over the past 35 years including obvious declines in the total area (by 42%) and number (by 51%) from 1980 to 2010 and a slight increase in the total lake area and number from 2010 to 2015. More than 235 lakes in the size class of 1–10 km$^2$ decreased to small lakes (area < 1 km$^2$), while 59 lakes covering 243.75 km$^2$ disappeared. Total reservoir area and number had continuous increases during the investigated 35 years, with an areal expansion of 54.9% from 919 km$^2$ to 1422 km$^2$, and a number increase by 65.3% from 78 to 129. The SEM revealed that the lake area in the NJW had a significant correlation with the mean annual precipitation (MAP), suggesting that the MAP decline clarified most of the lake shrinkage in the NJW. Furthermore, agricultural consumption of water had potential impacts on lake changes, suggested by the significant relationship between cropland area and lake area.

**Keywords:** lakes; reservoirs; Nenjiang watershed; climate change; Landsat

## 1. Introduction

Inland waterbodies (lakes and reservoirs) are closely related to human life due to their diverse ecosystem services [1,2]. Lakes and reservoirs in the drylands, although covering only a small proportion of the landscape, play irreplaceable roles in fragile environments and for local residents [3]. Meanwhile, they are sensitive to climate change and human disturbances [4]. Monitoring changes in lakes and reservoirs and investigating their driving forces are thus of great significance to sustainable water resource management and regional economic development.

Due to diverse driving forces, lakes around the world have experienced changes in both area and number during the past decades. Thaw and "breaching" of permafrost have caused a widespread decline in the Arctic lake number and area [5]. Because of warmer air temperatures, which allow

higher evaporation rates, as the world's largest surface freshwater system, the total area of the Great Lakes of North America experienced a continuous decrease from the 1980s to 2005 [6]. Under the background of global warming, glaciers in China's Tibetan Plateau (TP) showed a general retreat, while the glacial lakes expanded notably [7]. However, a drier climate and degraded permafrost have led to lake shrinkage in some basins of the TP [8]. Lakes over Mongolia experienced areal decline mostly related to a drier climate, while lakes in Inner Mongolia had notable shrinkage which was attributed mainly to human consumption of water resources, particularly coal mining [9]. Moreover, the lake area in the Jianghan Plain of China decreased dramatically from the 1950s to 1998, which was mainly caused by agricultural cultivation.

During the past 60 years, the number of reservoirs has increased notably, and there are about 50,000 large reservoirs nowadays around the world [10]. For example, China constructed nearly 45,000 reservoirs in the Yangtze River Basin to meet the large demand for water resources. By 2013, 98,000 reservoirs had been built in China [11]. Previous studies [12,13] indicated that most of large river systems and lakes in China were affected by reservoirs. Due to the construction of large reservoirs in upstream areas, water was impounded upstream, which seriously affected the water supply of lakes in the middle and lower reaches [14]. Therefore, it is necessary to track and analyze the changes in lakes and reservoirs at different scales and regions to form a scientific management response [15,16].

Optical images from different satellite sensors were widely used to monitor spatiotemporal variations of inland waterbodies at multiple geographic scales [17–22]. The Moderate Resolution Imaging Spectroradiometer (MODIS) data have been widely used to assess the water extent at daily to 16-day timescales despite a spatial resolution at 250 or 500 m [23]. Yet, areal changes of small lakes or reservoirs with irregular shapes were not accurately delineated due to the coarse resolution of source data. Other optical sensors, such as Quickbird (DigitalGlobe, Longmont, Colorado, USA) and IKONOS (DigitalGlobe, Longmont, Colorado, USA), provide finer images comparable to aerial photography for the extraction of lake or reservoir boundaries [24]. However, those data are limited in application at broad scale due to the high costs, narrow swath size, and so on [25,26]. Landsat series images have a fine spatial resolution (30 or 80 m) compared with previously mentioned data and have provided the longest temporal and spatial records for surface observations since their first launch in 1972 [27]. Consequently, Landsat imagery has been widely-used remote sensing data in examining changes in lakes or reservoirs.

As one of the third-largest waterbody clusters, an eco-environmentally fragile area, and an important base for grain production, the Nenjiang watershed (NJW) plays an important role in ecological conservation and national grain security in China [28]. However, the changes in lakes and reservoirs across the NJW during recent decades have rarely been examined and their correlations with climate change and human disturbances have rarely been quantitatively investigated. Doing so is critical to understanding the regional water cycle and sustainable water resource management. Therefore, this study aims to (1) employ Landsat 8 images to investigate the current status of lakes and reservoirs in the NJW, (2) evaluate changes in area and number of lakes and reservoirs using consistent Landsat images (i.e., 1980, 1990, 2000, 2010, and 2015), and (3) quantify the roles of climatic factors and artificial variables in driving lake changes.

## 2. Materials and Methods

### 2.1. Study Area

The NJW, covering an area of $297 \times 103$ km$^2$, is located in the core region of Northeast China (Figure 1), with latitudes from 44°1′48″ N to 51°42′1″ N and longitudes from 119°12′1″ E to 127°54′2″ E [28]. Obvious terrain variances can be found in this area with elevations ranging from 120 to 1740 m above sea level. The highest elevation is in the northwest at the Greater and Lesser Khingan Mountains, and the lowest elevation is in the southeast at the Songnen Plain. This is why the lakes and reservoirs are mainly observed in the southeast. The NJW is dominated by a temperate

semi-arid continental climate with an mean annual precipitation of around 470 mm and a mean annual air temperature of 4 °C [29]. Most precipitation (82%) occurs in months from June to September which is a critical growth period for vegetation in this region. As an important grain base in China, the NJW plays an critical role in promoting regional economic development and ensuring national food security [30,31]. Moreover, this region provides important habitats for threatened species and migratory waterfowls.

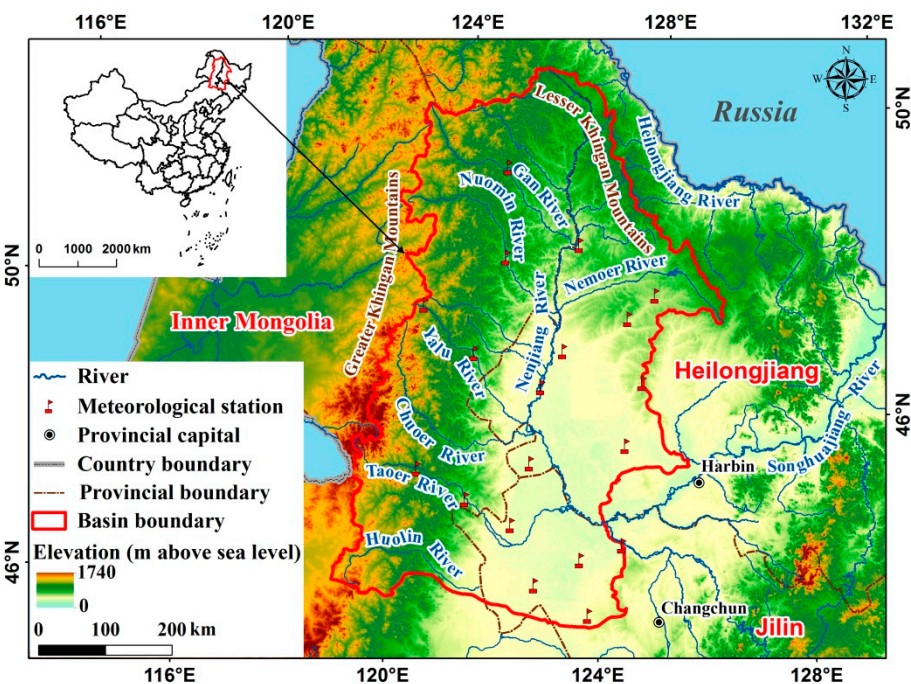

**Figure 1.** Geographic location and general situation of the Nenjiang watershed.

## 2.2. Data Source and Processing

### 2.2.1. Satellite Data

In this study, multi-temporal Landsat images from multispectral scanner (MSS), thematic mapper (TM), enhanced thematic mapper plus (ETM+), and operational land imager (OLI) sensors were acquired from the United States Geological Survey (USGS) to investigate the changes of lakes and reservoirs from 1980 to 2015. Specifically, we used 28, 27, 27, 29, and 26 scenes of images to extract the lakes and reservoirs for five dates of 1980, 1990, 2000, 2010, and 2015, respectively, (Figure 2). A total of 137 images acquired for months from June to September were used for the classification because the vegetation in this period is easier to be identified and the lakes and reservoirs have the largest amount. All of the images have a little cloud cover (less than 5%) and lakes/reservoirs on these images are clearly visible. In addition, in order to analyze the impact of floods on the area and the number of lakes and reservoirs in 1998 and 2013, we used 30 scenes of TM and 30 scenes of OLI images to extract the lakes and reservoirs in 1998 and 2013, respectively. Prior to image classification, data preprocessing including geometric, topographic, and radiometric corrections was performed for all the images using the ENVI 5.3 (Exelis Visual Information Solutions, Inc. Boulder, USA) software package. To ensure the data consistency, all images were re-projected to the 1984 WGS UTM zone 51N projection.

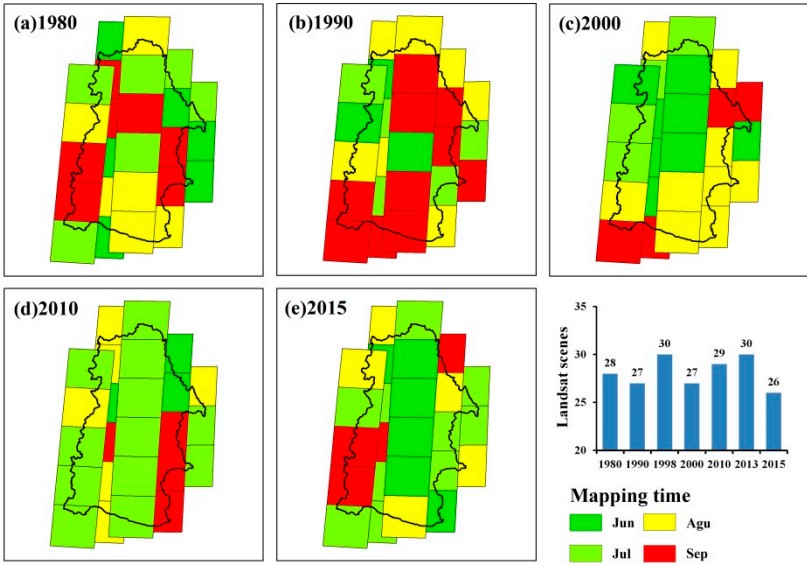

**Figure 2.** Landsat images selection in optimum months for different dates.

### 2.2.2. Meteorological Data

In order to analyze the relationships of lake/reservoir area with climatic factors, the daily climatic data, including extreme air temperature and precipitation, mean wind speed, and sunshine hours during 1980–2015, were collected from the meteorological records of the China Meteorological Data Service Center. Figure 1 shows the locations of the meteorological stations in the NJW. Spatial patterns of the mean annual air temperature (MAAT) and mean annual precipitation (MAP) were interpolated from those meteorological records using an Anusplin software considering the elevation differences [32]. We used the daily meteorological data and the Food and Agriculture Organization of the United Nations (FAO) Penman–Monteith model [8,33] to calculate the actual evapotranspiration (ET). ET is calculated as follows:

$$ET_0 = \frac{0.408\Delta(R_n - G) + \gamma \frac{900}{T+273} U_2(e_s - e_a)}{\Delta + \gamma(1 + 0.34U_2)} \tag{1}$$

$$ET = 9.78 + 0.0072 \times ET_0 \times PPT + 0.051 \times PPT \times LAI \tag{2}$$

where $ET_0$ is the potential evapotranspiration, $R_n$ is the net radiation, $G$ is the soil heat flux, $e_s$ is the saturation vapor pressure, $e_a$ is the actual vapor pressure, $e_s - e_a$ is the saturation vapor pressure deficit of the air, $T$ is the air temperature at 2 m height, $U_2$ is the wind speed at 2 m height, $\Delta$ is the slope of the saturation vapor pressure temperature relationship, $\gamma$ is the psychrometric constant, $PPT$ is precipitation, and the calculation of $LAI$ is referenced from the method of Lu et al. [34]. These parameters are directly calculated or derived from the average daily maximum and minimum temperature, the daily average temperature, the daily actual vapor pressure, the daily wind speed data, the actual duration of sunshine hours, the relative humidity data, and other empirical metrics.

In order to analyze the potential impact of climate change in the next 35 years (2015–2050) on regional lake area and number changes, we downloaded the Representative Concentration Pathways 5 (RCP 5) datasets from the China Agrosys Platform (http://stdown.agrivy.com). The future climate change data were generated based on the Fifth Generation Coupled Global Climate Model (CGCM 5) from the Canadian Centre for Climate Modeling and Analysis. The datasets constructed using original meteorological observations for each station included daily mean temperatures, daily precipitation, and daily potential evaporation. We converted daily mean temperature, daily precipitation, and daily potential evapotranspiration into annual mean temperature, annual mean precipitation, and annual potential evapotranspiration for our analysis.

### 2.2.3. Other Data

The areal data of the cropland area and gross domestic product (GDP) index collected from local statistical yearbook (http://www.stats.gov.cn).

### *2.3. Data Analysis*

#### 2.3.1. Extracting Lakes and Reservoirs

We focused on lakes and reservoirs greater than 1 km$^2$ in terms of the 30 and 80 m resolutions of the satellite images [35]. An object-based image analysis (OBIA) was used rather than the traditional pixel-based classification method, because OBIA classifies objects instead of individual pixels [36–38]. In the process of OBIA classification, the spectrum, spatial information, texture, and geometric features characterized by remote sensing images were fully utilized. The lake and reservoir extents of the NJW in 1980, 1990, 1998, 2000, 2010, 2013, and 2015 were extracted in eCognition Developer 8.6 [14,39]. The normalized difference water index (*NDWI*) is the most popular used index for automated inland waterbodies delineation [35,40,41]. In addition, we tested different versions of *NDWI* (*mNDWI*, *NDRW*, and *NDWI*) [21,42] and found that the selected *NDWI* is more effective in our study area. Therefore, *NDWI* was applied to extract lakes and reservoirs, which was defined as:

$$NDWI = (Green - NIR)/(Green + NIR) \qquad (3)$$

where *Green* and *NIR* represent the reflectance of the green and Near Infrared (*NIR*) bands, respectively [25]. The data processing steps for the lake and reservoir inventory are shown in Figure 3. The extraction of lakes and reservoirs consisted of three major steps: image multi-resolution segmentation, *NDWI* threshold testing, and classification rule designing. By defining the length/width index and the rectangular fit index built into the software, rivers and artificial ponds were removed. Then, we used visual interpretation to extract the reservoirs.

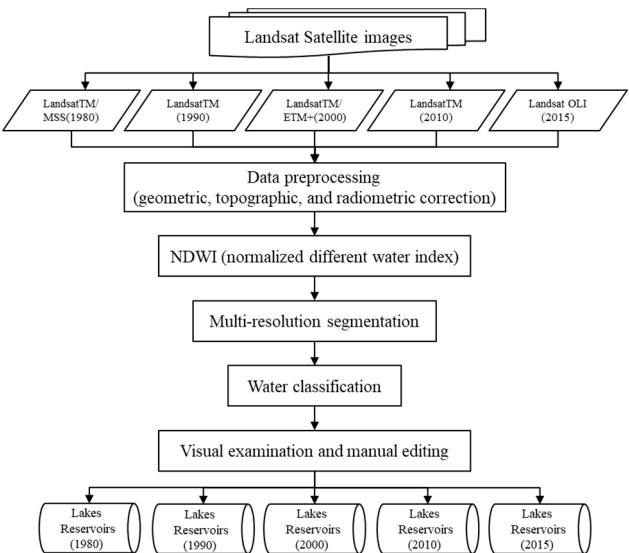

**Figure 3.** Flowchart for the lake and reservoir mapping process.

To validate the accuracy of extracted lakes and reservoirs, we calculated error matrices based on 1492 validation samples selected from Google Earth and Landsat sensors. The detailed number of validation samples for the five dates is given in Table 1. The overall accuracies of the lake and reservoir classification for the five dates were respectively evaluated and the standard errors bars (uncertainties) were estimated [43]. The overall accuracies of lakes and reservoirs were larger than 90%, and the kappa coefficients were larger than 0.87.

**Table 1.** Collected validation samples for the five dates.

| Date | 1980 | 1990 | 2000 | 2010 | 2015 |
|---|---|---|---|---|---|
| Sources | Google Earth Image/MSS | TM | TM/ETM+ | TM | OLI |
| Lakes | 126 | 158 | 170 | 244 | 256 |
| Reservoirs | 38 | 47 | 50 | 62 | 65 |
| Non-water bodies | 35 | 40 | 79 | 65 | 57 |
| Total Samples | 199 | 245 | 299 | 371 | 378 |

MSS—multispectral scanner, TM—thematic mapper, ETM+—enhanced thematic mapper plus, OLI—operational land imager.

### 2.3.2. Temporal Analysis of Lakes and Reservoirs

To fully understand the changes of lakes and reservoirs of different sizes, the lakes and reservoirs were categorized into four classes: 1–10 km$^2$, 10–50 km$^2$, 50–100 km$^2$, and >100 km$^2$. In order to examine the areal change rate of the lake or reservoir in different periods, an indicator of the lake or reservoir area dynamic degree, as shown Equation (4), was used to analyze their changes [44,45].

$$K = \frac{U_b - U_a}{U_a} \times \frac{1}{T} \times 100\% \tag{4}$$

where $K$ is the dynamic indicator for the lake or reservoir area; $U_a$ and $U_b$ are the area of the lake and reservoir at the start date and end date, respectively; and $T$ is the time scale under consideration.

### 2.3.3. Assessing the Roles of Climatic Factors and Anthropogenic Causes in Lake Changes

The roles of climatic factors and artificial variables in changes of lakes during the study period were quantified using structural equation modeling (SEM) by the AMOS 22 software (IBM, Armonk, NY, USA) [39]. This paper analyzed the changes in MAAT, MAP, and ET to examine the influences of climate change on lake changes. The statistical data of cropland area and GDP were selected to investigate the impacts of artificial variables on lake changes [46,47]. Table 2 illustrates the optimum values for these indicators necessary for the SEM.

**Table 2.** Measures used to test the goodness of model.

| Measure | Optimum Values | Reference |
|---|---|---|
| RMSEA (root mean square error of approximation) | Less than 0.08 | Li et al. (2019) [39] |
| $\lambda^2$/df (chi-square/degree of freedom) | Less than 3 | James (2007) [48] |
| GFI (goodness of fit index) | 0.90 and above | Melucci et al. (2019) [49] |
| CFI (comparative fit index) | 0.90 and above | David et al. (2000) [50] |

## 3. Results

### 3.1. Spatial Pattern of Lakes and Reservoirs in 2015

Figure 4 shows the lake and reservoir distribution in 2015 across the NJW. Lakes and reservoirs were identified dominantly in the southeastern part of the NJW. A total of 233 lakes (area ≥ 1 km$^2$) covering an area of 2110 ± 53 km$^2$ in 2015 were extracted from satellite images. Most of the lakes (89.4%) had an area of smaller than 10 km$^2$. There were three lakes with area greater than 100 km$^2$, and their accumulated area accounted for approximately 33.8% of the total lake area in the NJW. The Chagan Lake was the largest lake with an estimated area of 292 km$^2$.

There were 129 reservoirs (area ≥ 1 km$^2$) in the NJW with a total area of 1422 ± 56 km$^2$ in 2015. Of these, 108 reservoirs had an area smaller than 10 km$^2$. There were two reservoirs with an area larger than 100 km$^2$, with the accumulated area accounting for approximately 41.5% of the total reservoir area in the NJW. The Nierji Reservoir was the largest reservoir with an estimated area of 360 km$^2$.

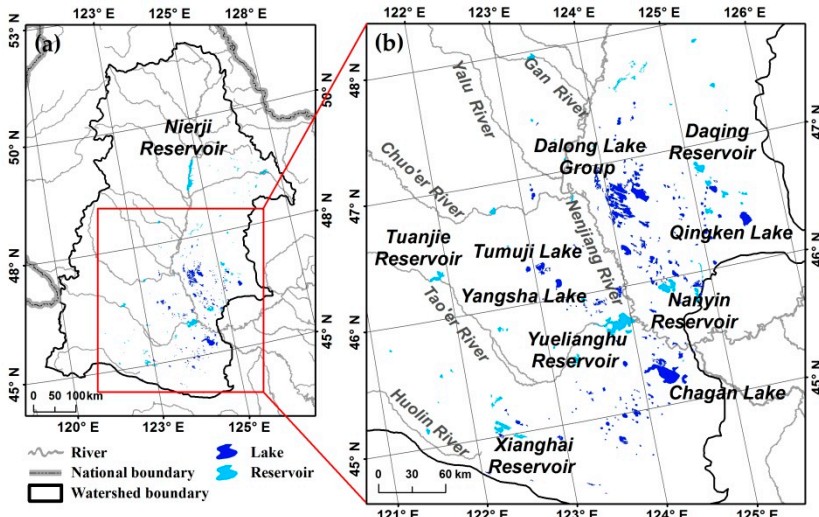

**Figure 4.** Spatial distribution of lakes and reservoirs in the Nenjiang watershed (NJW) in 2015.

### 3.2. Temporal Changes of Lakes and Reservoirs from 1980 to 2015

Figure 5 suggests that dramatic changes in the total area and number of lakes and reservoirs over the NJW occurred from 1980 to 2015. During the observed 35 years, the total lake area decreased by 38.7% from $3440 \pm 72$ km$^2$ to $2110 \pm 53$ km$^2$, whilst the total number of lakes decreased by 233 from the original 468 in 1980. Specifically, lake changes had significant fluctuations over the past 35 years, including obvious declines in total area (42%) and number (51%) from 1980 to 2010 and slight increases in the total lake area and number from 2010 to 2015. Reservoirs in the NJW experienced continuous expansion during 1980–2015. The total number of reservoirs increased from 78 to 129 with the total area expansion being 55%, from $919 \pm 53$ km$^2$ to $1422 \pm 56$ km$^2$.

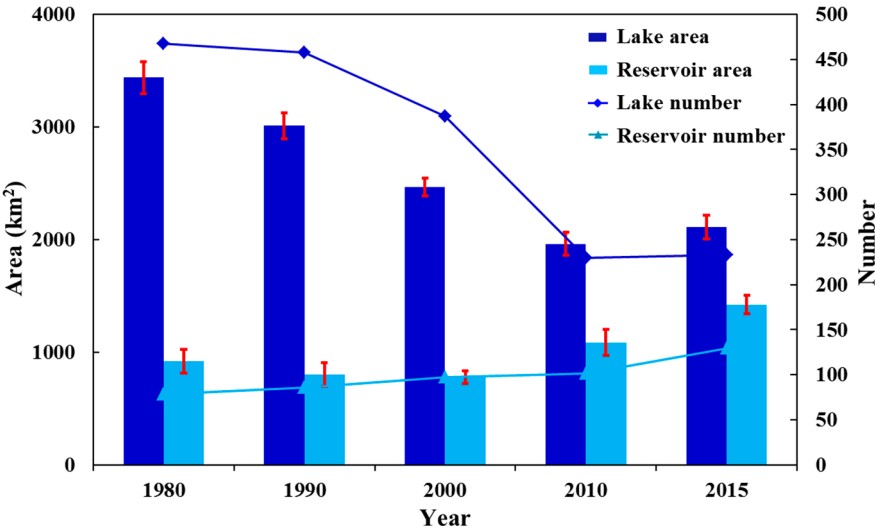

**Figure 5.** Decadal variations in the total area and number of lakes and reservoirs (area $\geq 1$ km$^2$) in the NJW from 1980 to 2015. The error bars represent the 95% confidence level.

#### 3.2.1. Lake Changes

For a further understanding of the temporal changes of lakes, the spatial heterogeneity of lake area changes across the NJW was investigated (Figure 6). Changes in the lake area presented clear variations. During the first period, 1980–1990, most of the expanded lakes were mainly distributed in the southeastern part of the NJW, while the lakes that shrunk were mainly distributed in the central

part (Figure 6a). A rapid shrinkage of the total lake area and a decline in lake number occurred during 1990–2000. The larger lake shrinkage occurred in the southern NJW (Figure 6b), while lakes in the Chagan Lake, Yangsha Lake, and Tumuji Lake zones exhibited expanding trends during this period. During 2000–2010, lakes in the NJW with large areal loss were distributed mainly in the Dalong Lake zones, while expanded lakes were identified mainly at the intersection point between the Nenjiang River and Taoerhe River (Figure 6c). During 2010–2015, it is noteworthy that the characteristic of lake expansion was relatively obvious in the NJW compared to the other periods. Lakes with shrinkage were mainly distributed in the southeastern part, whereas lake expansions mostly occurred in the eastern part (Figure 6d).

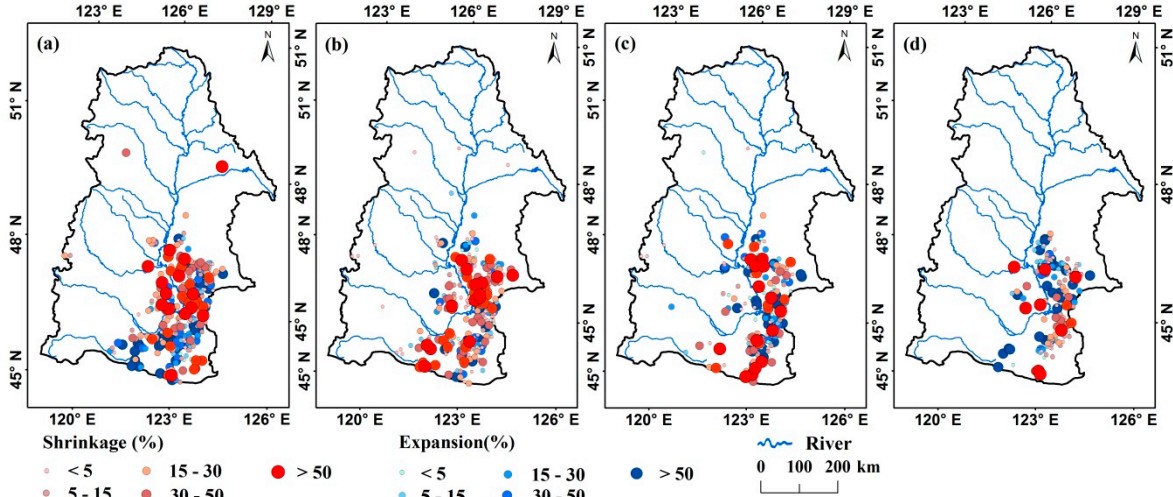

**Figure 6.** Spatial variations of lake changes in the Nenjiang watershed during different periods. Red and blue dots present lake shrinkage and expansion, respectively, while the dot extent denotes change proportion in the corresponding period. (**a**–**d**) represent the periods 1980–1990, 1990–2000, 2000–2010, and 2010–2015, respectively.

The detailed lake changes in each class are shown in Table 3. From 1980 to 1990, the largest areal decline of lakes was observed for the 50–100 km$^2$ size class. Interestingly, the Dalong Lake, which was in the size class of 50–100 km$^2$, separated into five lakes in the size class of 10–50 km$^2$ during this period. Specifically, 10 lakes with a size of 1–10 km$^2$ covering a total area of 34 km$^2$ in the central region disappeared in this period. During 1990–2000, lake shrinkage occurred with the total lake area declining by 18%, and 53 lakes (area ≤ 50 km$^2$) vanished during this decade. During 2000–2010, both the total number and area of lakes in the size class of 50–100 km$^2$ increased, while both the total number and area of lakes in other size classes decreased. From 2010 to 2015, both the total area and number of lakes in the size classes 1–10, 10–50, and 50–100 km$^2$, increased.

**Table 3.** Lake number, area, and changes in different classes between 1980 and 2015.

| Year or Period | Area Classes (km$^2$) | | | | Total |
| --- | --- | --- | --- | --- | --- |
| | 1–10 | 10–50 | 50–100 | >100 | |
| **Number of lakes** | | | | | |
| 1980 | 419 | 43 | 3 | 3 | 468 |
| 1990 | 414 | 40 | 1 | 3 | 458 |
| 2000 | 343 | 39 | 0 | 4 | 386 |
| 2010 | 198 | 27 | 2 | 3 | 230 |
| 2015 | 199 | 29 | 2 | 3 | 233 |
| **Change in number (%)** | | | | | |
| 1980–1990 | −1 | −7 | −67 | 0 | −2 |
| 1990–2000 | −17 | −3 | 0 | 33 | −15 |
| 2000–2010 | −42 | −31 | 100 | −25 | −41 |
| 2010–2015 | 0 | 7 | 0 | 0 | 1 |
| 1980–2015 | −53 | −33 | −33 | 0 | −50 |
| **Lake area (km$^2$)** | | | | | |
| 1980 | 1369 ± 11 | 916 ± 17 | 197 ± 20 | 958 ± 24 | 3440 ± 72 |
| 1990 | 1145 ± 10 | 884 ± 15 | 81 ± 17 | 900 ± 17 | 3010 ± 59 |
| 2000 | 934 ± 8 | 729 ± 19 | 0.0 | 804 ± 13 | 2467 ± 40 |
| 2010 | 548 ± 7 | 535 ± 16 | 111 ± 17 | 767 ± 13 | 1961 ± 53 |
| 2015 | 595 ± 7 | 656 ± 14 | 145 ± 15 | 714 ± 17 | 2110 ± 53 |
| **Change in area (%)** | | | | | |
| 1980–1990 | −16 ** | −4 ** | −59 ** | −6.0 ** | −13 ** |
| 1990–2000 | −18 ** | −18 ** | −100 ** | −10.7 ** | −18 ** |
| 2000–2010 | −41 ** | −27 ** | 0 | −4.5 ** | −20 ** |
| 2010–2015 | 9 ** | 23 ** | 30 ** | −7.0 ** | 8 ** |
| 1980–2015 | −57 ** | −28 ** | −26 ** | −25.5 ** | −39 ** |

Note: * denotes change at a significant level of 0.05, ** denotes significant at 0.01.

### 3.2.2. Reservoir Changes

Figure 7 shows the spatiotemporal patterns of reservoir changes in the NJW during different periods. During 1980–1990, the reservoirs that shrunk were distributed mainly in the central and northeast parts, whereas the expanded reservoirs were mainly distributed in the southern and eastern parts of the NJW (Figure 7a). During 1990–2000, the expanded reservoirs were mainly distributed in the northeast and southern parts, whereas the reservoirs that shrunk were mainly distributed in the central and southern parts of the NJW (Figure 7b). During 2000–2010, significant reservoir expansions were identified across the NJW, while the larger reservoirs that shrunk were mainly distributed in the southern part of the NJW (Figure 7c). After 2010, the reservoirs that shrunk were mainly distributed in the southern and eastern parts. Larger expanded reservoirs were mainly identified in the northern part and the Taoer River basin (Figure 7d).

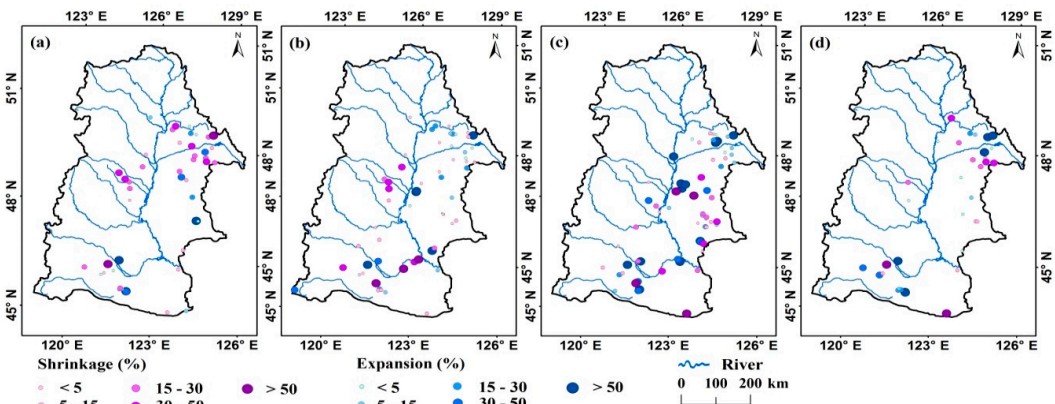

**Figure 7.** Spatial distribution of reservoir changes in the Nenjiang watershed during different periods. Purple and blue dots present reservoir shrinkage and expansion, respectively, while the dot extent denotes change proportion in the corresponding period. (**a**–**d**) represent the periods 1980–1990, 1990–2000, 2000–2010, and 2010–2015, respectively.

During the period of 1980–1990, the total area of reservoirs in size classes of 10–50 km$^2$ and 50–100 km$^2$ increased by 121%, with an areal increase from 149 ± 10 to 328 ± 16 km$^2$, and by 133% from 52 ± 14 to 121 ± 17 km$^2$, respectively. The total areas of reservoirs in the size classes of 1–10 km$^2$ and >100 km$^2$ decreased from 205 ± 9 to 200 ± 6 km$^2$ (−2%) and from 513 ± 20 to 151 ± 14 km$^2$ (−71%), respectively (Table 4). During 1990–2000, the total areas of reservoirs in the size classes of 1–10 km$^2$ and 50–100 km$^2$ increased by 23%, with an areal increase from 200 ± 6 to 247 ± 5 km$^2$, and by 123% from 121 ± 17 to 270 ± 15 km$^2$, respectively. The total area of reservoirs in the size classes of 10–50 km$^2$ decreased from 328 ± 16 to 263 ± 8 km$^2$ (−20%). During the period of 2000–2010, the total area of reservoirs increased from 780 ± 28 to 1086 ± 58 km$^2$. The total areas of reservoirs in the size classes of 1–10 km$^2$ and 10–50 km$^2$ increased by 30% and 24%, respectively. The total area of reservoirs in the size class of 50–100 km$^2$ decreased by 50%. During the period of 2010–2015, the total reservoir area increased by 31% with an increase in area from 1086 ± 58 to 1422 ± 56 km$^2$. Detailed reservoir changes for these different classes are shown in Table 4.

**Table 4.** Reservoir number, area, and changes in different classes between 1980 and 2015.

| | Year or Period | Area Classes (km$^2$) | | | | Total |
|---|---|---|---|---|---|---|
| | | **1–10** | **10–50** | **50–100** | **>100** | |
| **Number of reservoirs** | 1980 | 69 | 6 | 1 | 2 | 78 |
| | 1990 | 68 | 15 | 2 | 1 | 86 |
| | 2000 | 73 | 19 | 4 | 0 | 96 |
| | 2010 | 84 | 13 | 2 | 2 | 101 |
| | 2015 | 108 | 17 | 2 | 2 | 129 |
| **Change in number (%)** | 1980–1990 | −1 | 150 | 100 | −50 | 10 |
| | 1990–2000 | 7 | 27 | 100 | 0 | 13 |
| | 2000–2010 | 15 | −32 | −50 | 100 | 4 |
| | 2010–2015 | 29 | 31 | 0 | 0 | 28 |
| | 1980–2015 | 57 | 183 | 100 | 0 | 65 |
| **Reservoir area (km$^2$)** | 1980 | 205 ± 9 | 149 ± 10 | 52 ± 14 | 513 ± 20 | 919 ± 53 |
| | 1990 | 200 ± 6 | 328 ± 16 | 121 ± 17 | 151 ± 14 | 800 ± 53 |
| | 2000 | 247 ± 5 | 263 ± 8 | 270 ± 15 | 0 | 780 ± 28 |
| | 2010 | 320 ± 6 | 324 ± 16 | 134 ± 17 | 308 ± 19 | 1086 ± 58 |
| | 2015 | 259 ± 4 | 397 ± 16 | 175 ± 18 | 591 ± 18 | 1422 ± 56 |
| **Change in area (%)** | 1980–1990 | −2 | 121 [**] | 133 [**] | −71 [**] | −13 [**] |
| | 1990–2000 | 23 [**] | −20 [**] | 123 [**] | −100 [**] | −3 |
| | 2000–2010 | 30 [**] | 24 [**] | −50 [**] | 0 | 39 [**] |
| | 2010–2015 | −19 [**] | 23 [**] | 31 [**] | 92 [**] | 31 [**] |
| | 1980–2015 | 27 [**] | 167 [**] | 237 [**] | 15 [**] | 55 [**] |

Note: * denotes change at a significant level of 0.05, ** denotes significant at 0.01.

### 3.3. Roles of Climatic Factors and Artificial Variables in Driving Lake Changes

As shown in Figure 8, the model passed the reliability test, convergent validity test, and discriminant validity test. The dominant climatic factor that influenced lake changes was the MAP ($\beta = 0.66$, $p < 0.001$), followed by the ET ($\beta = 0.39$, $p < 0.05$) and MAAT ($\beta = 0.15$, $p < 0.1$). Agricultural consumption of water had a significant effect on lake changes, suggested by the significant relationship between cropland area and lake area ($\beta = 0.17$, $p < 0.1$).

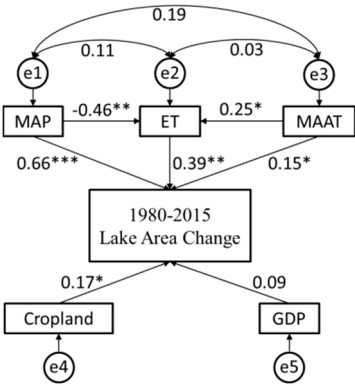

**Figure 8.** Structural model and path coefficient. Arrows show the effect path and direction. Numbers adjacent to arrows are path coefficients, (β). The path coefficients (β) characterize the extent of the effect of the examined variable on the lake changes, while the larger value indicates a strong positive or negative effect. P represents a significant level. * represents *p*-value < 0.1, ** represents *p*-value < 0.05, and *** represents *p*-value < 0.01. MAP denotes the mean annual precipitation, MAAT represents the mean annual air temperature, ET represents the evapotranspiration, and GDP represents the gross domestic product. The "e" represents the error (residual term) of path analysis of observed variables in the structural model. The numbers on the arrows are the values of the standardized regression weights of the model.

## 4. Discussion

Lake and reservoir mapping is affected by the resolution of the used satellite data [26]. We focused only on lakes and reservoirs with area greater than 1 km$^2$ to reduce as much as possible the errors induced by a coarse resolution of 30 and 80 m. Although some images out of the optimal season were used, these data mainly covered the Greater and Lesser Khingan Mountains where few lakes and reservoirs were identified. This did not yield large uncertainties in our analysis. This study integrated OBIA and visual interpretation instead of automatic classification to extract lakes and reservoirs, which ensured data accuracy and effective analysis.

Both climate change and human activities contributed to the changes of lakes in the NJW. On the one hand, the climate over the study area is a semi-arid continental climate. Water supply and output for these lakes dominated with precipitation and evapotranspiration, respectively. SEM analysis revealed that the lake shrinkage in the NJW had a significant correlation with the MAP (β = 0.66, *p* < 0.001) (Figure 8), followed by the ET (β = 0.39, *p* < 0.05), and the MAAT (β = 0.15, *p* < 0.1). This suggests that precipitation had a significant statistical relationship with lake area and potentially had the largest impacts on the lake area. During the investigated 35 years, a warmer climate was identified for the NJW with a significant increase of MAAT (*p* < 0.05) (Figure 9a). A reduced water supply from precipitation and output by increased ET characterized most of the lake shrinkages in the NJW. Specifically, lakes in the NJW showed areal changes with a decrease from 1980 to 2010 and then an increase from 2010 to 2015 (Figure 5), which is consistent with the changed trend of MAP during the 35 years (Figure 9a). The MAP could be regarded as the main climatic factor to explain the lake shrinkage in the NJW from 1980 to 2015 in terms of its decline.

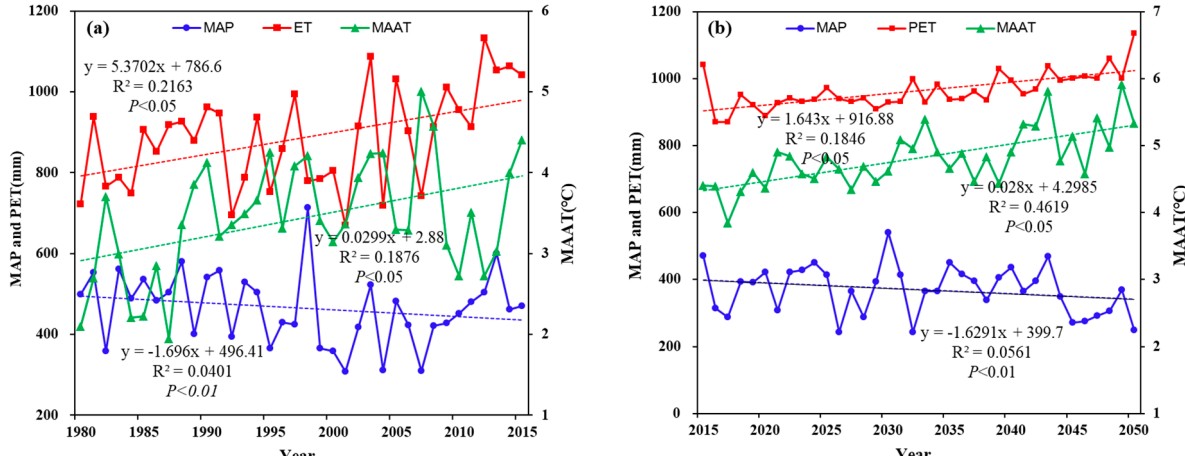

**Figure 9.** Past changes and future projections of climatic factors in the NJW. (**a**) Changes of mean annual air temperature (MAAT), mean annual precipitation (MAP), and evapotranspiration (ET) from 1980 to 2015 and (**b**) changes of future MAAT, MAP, and PET from 2015 to 2050.

Future projections of climate change using the Intergovernmental Panel on Climate Change (IPCC) models (RCP 5) indicate that the warming and drying trend will continue in the NJW (Figure 9b). Both the MAAT and PET show increasing trends with a rate of 0.03 °C yr$^{-1}$ and 1.64 mm yr$^{-1}$ ($p < 0.05$), while the MAP exhibits a significant decline with a rate of $-1.63$ mm yr$^{-1}$ ($p < 0.01$). If this trend continues, the total area of lakes in the NJW may continue to decrease, and some small lakes will disappear.

This study found that flood events can markedly affect the lake area and number, especially for the small lakes (1–10 km$^2$). According to the hydrological records, the most serious floods during the recent century occurred in 1998 and 2013 [51]. Compared with the total lake area in 2010 (normal flow year), the total lake area in the NJW respectively increased by 2946 km$^2$ in 1998 and 895 km$^2$ in 2013 during the flooding events. It is clear that extreme precipitation events (floods) have accelerated the expansion of the total lake area (Figure 10). In arid and semi-arid regions, a multidimensional view on the prevention and exploitation of floods is required. Lakes and reservoirs have huge water storage capacity, and thus they could serve as hydrological buffers to prevent floods and provide irrigation water for sustainable agricultural development [10].

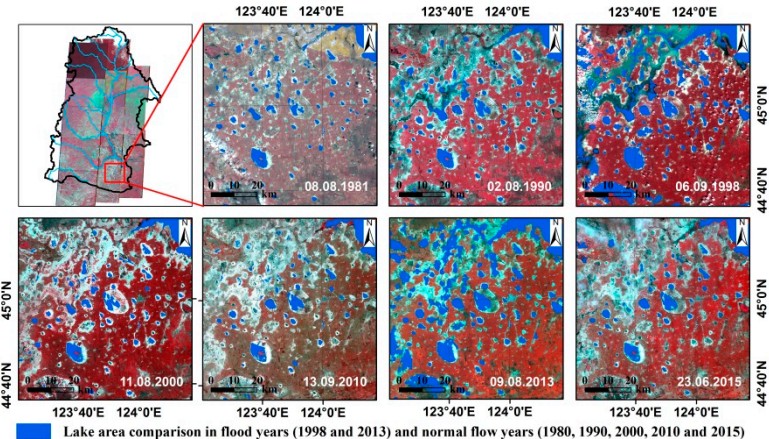

**Figure 10.** Increased lake area due to floods in 1998 and 2013.

On the other hand, the study area is an important grain production base in China. Agricultural demand for water had potential impacts on lakes. Besides climate change, human activities have imposed marked impacts on lakes and reservoirs [52]. While our study revealed that climatic factors drive striking lake and reservoir changes, human-induced changes should be responded to as quickly

as possible. In this study, a clear correlation (β = 0.17, *p* < 0.1) was observed between lake area and crop area (Figure 8). Lake shrinkage caused by artificial variables is mainly attributed to the agricultural water consumption in the NJW [50,53]. Due to the construction of large reservoirs upstream water was therefore impounded upstream, which seriously affected the water supply of lakes in the middle and lower reaches. In particular, the largest reservoir, Nierji Reservoir, constructed over the upstream area of the Nenjiang River exerted a marked influence on downstream lakes. Due to reclamation from lakes and agricultural development during the past 35 years, some lakes in the size class of 1–10 km² decreased to small lakes with areas smaller than 1 km². For example, a large area of shallow waters was reclaimed for planting rice (Figure 11). Since 1990, the area of paddy fields in the NJW has increased significantly by 90.26 km² from 3.46 to 93.73 km². The total cropland area increased from 459.42 to 948.18 km². With the population increase and agricultural land expansion in this region, more and more open water resources have been applied to agricultural irrigation. It is evident that the human impacts on lake and reservoir changes could not be ignored.

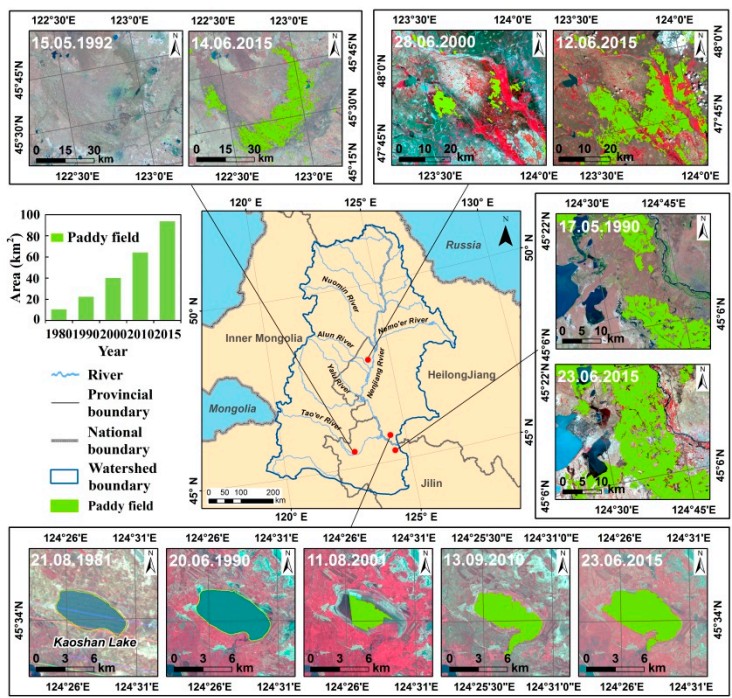

**Figure 11.** Image examples for lakes and reservoirs affected by the expanded paddy fields.

Lake shrinkage and disappearance lead to ecological and environmental degradation, such as aggravating the degree of sandstorms and desertification and reducing the number of wild animals [4,54]. To optimize water distribution programs through scientific water conservancy projects is very important. Therefore, appropriate measures need to be implemented by managers to further reduce the decline of lake areas [55], such as enhancing the drainage capacity of reservoirs in drought years but the storage capacity of reservoirs in flood years. In addition, we should control the expansion of paddy fields to relieve water stress and guarantee regional sustainable development in the NJW [56].

## 5. Conclusions

In this paper, we established a multi-temporal dataset of lakes and reservoirs in the NJW using long time-series Landsat images from 1980 to 2015 to document their changes on a decadal scale and quantified the contribution degree of MAAT, MAP, ET, cropland, and GDP to change in lake area. A notable decline in the total lake area by 1330 km² in the period of 1980 to 2015 was identified, while the lake number decreased contemporaneously. In contrast to lake shrinkage, the total area and number of reservoirs in the NJW experienced continuous increases. We identified 51 newborn reservoirs with

total area of 504 km² in 2015 compared to 1980. SEM analysis revealed that decrease of the MAP is the dominant factor driving the changes of lakes, followed by ET and MAAT, especially for those of small size. Furthermore, the human impacts on lake and reservoir changes could not be ignored. Timely and appropriate policies and measures are required to reduce lake shrinkage and respond to the degraded environment. The results and analysis in this study are expected to provide guidance for sustainable management of water resource in the NJW.

**Author Contributions:** B.D. and D.M. designed the analytical framework of this study. B.D. performed the data analysis and drafted the manuscript. H.L. and H.X. provided methodological advice. D.M. and Z.W. made major revisions of the manuscript. All authors have read and agreed to the published version of the manuscript.

**Funding:** This study was jointly funded by the National Key Research and Development Program of China (2019YFA0607100, 2016YFC0500408, 2016YFC0500201, and 2016YFA0602301), the National Natural Science Foundation of China (41771383), and the funding from Youth Innovation Promotion Association, Chinese Academy of Sciences (2017277, 2012178) and National Earth System Science Data Center of China (http://northeast.geodata.cn).

**Acknowledgments:** We appreciate the facility that make Landsat 8 OLI images accessible through the USGS. We thank the three anonymous reviewers for the constructive comments and suggestions, which help improve the quality of this manuscript.

**Conflicts of Interest:** The authors declare no conflict of interest.

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
