# Peer review of "Tracking Lake and Reservoir Changes in the Nenjiang Watershed, Northeast China: Patterns, Trends, and Drivers"

_water, doi:10.3390/w12041108_

Round 1

Reviewer 1 Report

Water-761576

This paper addresses an interesting and up-to-date subject. Although the manuscript does not have high novelty, it is well written and presents an interesting case study with much needed sustainable management in the future.

The manuscript is very polished, the structure is good, figures are well established, English very good and a very nice Discussions chapter. I don’t know if the manuscript was previously revised or not, because I have no new recommendations or faults with this article, and it is the first time this happened to me.

I would like to congratulate the authors for the effort and scope of the article. It presents an interesting topic and has good readability. Also, I appreciated the fact that it was concise, with perfect balance of introduction, M&M, results and discussions.

Author Response

[Comment]: This paper addresses an interesting and up-to-date subject. Although the manuscript does not have high novelty, it is well written and presents an interesting case study with much needed sustainable management in the future.

The manuscript is very polished, the structure is good, figures are well established, English very good and a very nice Discussions chapter. I don’t know if the manuscript was previously revised or not, because I have no new recommendations or faults with this article, and it is the first time this happened to me.

I would like to congratulate the authors for the effort and scope of the article. It presents an interesting topic and has good readability. Also, I appreciated the fact that it was concise, with perfect balance of introduction, M&M, results and discussions.

Response: We appreciate the comments and thank you for recognizing the contribution of this study.

Reviewer 2 Report

This manuscript presents an interesting study by evaluating a multi-temporal dataset for lake and reservoir with area more than 1 km2 changes in the Nenjiang watershed, China, using an object-oriented image classification method and Landsat series images from 1980 to 2015 at period of 10 years. The results indicated that lakes have experienced significant changes with fluctuations over the past 35 years with declines in the total area of 42% and number of 51%, while the reservoir areas and number had continuous increases during the investigated years. The manuscript presents figures and tables which are supported the findings and conclusions of this study. In general, the manuscript is organized well and there are some parts that need to be clarified and reviewed by Professional English Editor. I did not observe any scientific lack in this manuscript and it is qualify for publication after considering some minor comments that listed below and that could improve the manuscript:

    • Abstract: the abstract should be rewritten and reviewed by professional English editor. I found multi-issues in this section. For instant, Line 28, the authors should be defined some parameters such as what are MAP, Beta, P refer to? Are Beta and P statistical parameters? the kind of information should be clarified
    • Introduction, Line 70: Is the term (30/80) mean that 30 to 80 m? Please, explain.
    • 2.1. Study area:
      • Line 87: The coordinates should be presented in degree, minutes, and seconds.
      • Line 93: Usually, in scientific written style, the sentence is not start with numerical number. Please, revise “82%” to “Eighty two percent”.
      • Figure 1: The elevation in legend should be adjusted from “m” to “m above sea level”  
    • 2.2.1. Satellite data, Line 109: Again more parameters are not defined (TM and OLI).
    • 3. Results:
      • 3.2.1. Lake changes, Figure 6: The caption of Figure 6 is not clear. The reader is not able to identify the shrinkage %, or Expansion%, from the caption. The authors should provide more information to identify the parameters without reading the text.

  • 3.2.2. Reservoir changes, Table 4, and lines 278-281: Please, define P and Beta.

Author Response

[Comment]: This manuscript presents an interesting study by evaluating a multi-temporal dataset for lake and reservoir with area more than 1 km2 changes in the Nenjiang watershed, China, using an object-oriented image classification method and Landsat series images from 1980 to 2015 at period of 10 years. The results indicated that lakes have experienced significant changes with fluctuations over the past 35 years with declines in the total area of 42% and number of 51%, while the reservoir areas and number had continuous increases during the investigated years. The manuscript presents figures and tables which are supported the findings and conclusions of this study. In general, the manuscript is organized well and there are some parts that need to be clarified and reviewed by Professional English Editor. I did not observe any scientific lack in this manuscript and it is qualify for publication after considering some minor comments that listed below and that could improve the manuscript:

Response:We thank the reviewer very much for their constructive comments and suggestions on our manuscript. We have carefully considered all of the comments. Point-by-point responses to each comment are as follows:

2.1 [Comment]: Abstract: the abstract should be rewritten and reviewed by professional English editor. I found multi-issues in this section. For instant, Line 28, the authors should be defined some parameters such as what are MAP, Beta, P refer to? Are Beta and P statistical parameters? the kind of information should be clarified.

Response: Thanks for this comment. As suggested, we checked the abstract and added the full name of MAP in the Abstract. Added texts read as below.

The term “MAP” has been replaced with “mean annual precipitation (MAP)” in line 28.

In this study, structural equation models (SEM) reveals the effects of different factors on lake changes. The path coefficients (β) characterize the effects extent of the examined variable on the lake changes, while the larger value indicates a strong positive or negative effects. P represents a significant level.

The reviewer 3 recommend that it is not necessary to give the confidence intervals of the measurements or the parameters of statistical significance in the abstract.

Considering both the comments from two reviewers, we removed Beta and P from the revised manuscript and added their descriptions in the caption of Figure 8.

2.2 [Comment]: Introduction, Line 70: Is the term (30/80) mean that 30 to 80 m? Please, explain.

Response: We thank the reviewer for this comment. We replaced “30/80” with “30 or 80”.

2.3 [Comment]: Line 87: The coordinates should be presented in degree, minutes, and seconds.

Response: We thank the reviewer for this comment. As suggested, we added details for the coordinates in the revised manuscript. The revised texts read as below.

“with latitudes from 44°1′48″N to 51°42′1″N and longitudes from 119°12′1″E to 127°54′2″E in line 87.

2.4 [Comment]: Line 93: Usually, in scientific written style, the sentence is not start with numerical number. Please, revise “82%” to “Eighty two percent”.

Response: We thank the reviewer for this comment. We have revised this sentence as follows:

“There are 82% of precipitation occurring in months from June to September which is a critical growth period for vegetation in this region.”

2.5 [Comment]: Figure 1: The elevation in legend should be adjusted from “m” to “m above sea level”.

Response: We thank the reviewer for this comment. As suggested, we changed the legend for Figure 1.

2.6 [Comment]: 2.2.1. Satellite data, Line 109: Again more parameters are not defined (TM and OLI).

Response: Thanks for this comment. We have added full names for TM and OLI.

“Thematic Mapper (TM)”, “Operational Land Imager (OLI)”

2.7 [Comment]: 3.2.1. Lake changes, Figure 6: The caption of Figure 6 is not clear. The reader is not able to identify the shrinkage %, or Expansion%, from the caption. The authors should provide more information to identify the parameters without reading the text.

Response: We apologize for the lack of clarity on the caption of Figure 6. Following your suggestion, we added two sentences to clarify it. The added texts read as follows.

“Red and blue dots present lake shrinkage and expansion, respectively, while the dot extent denotes change proportion in the corresponding period.”

The related clarification was also added for caption of Figure 7.

2.8 [Comment]: 3.2.2. Reservoir changes, Table 4, and lines 278-281: Please, define P and Beta.

Response: As suggested, descriptions for Beta and p have been added in the revised manuscript.

“The path coefficients (β) characterize the effects extent of the examined variable on the lake changes, while the larger value indicates a strong positive or negative effects. P represents a significant level.”

“Notes: The * denotes change at a significant level of 0.05, ** denotes significant at 0.01.”

Reviewer 3 Report

The modifications suggested in the previous review have been incorporated into the manuscript. Only a few small details remain in the abstract and in the bibliography that need to be corrected.

In the abstract it is not necessary to give the confidence intervals of the measurements of the areas or the parameters of statistical significance. The abbreviation MAP is used and has not been previously defined. The final sentence is a speculative statement and can also be deleted.

In the references list, delete some dots:

Line 386, delete dot in Science. 2005

Line 404, 426, 466, delete dot in Water. 2015

Line 451, delete dot in One. 2012

Line 476, delete dot in Sensors. 2008

Line 486, delete dot in Systems. 2000

Line 500, delete dot in Procedia. 2010

Author Response

3.1 [Comment]: The modifications suggested in the previous review have been incorporated into the manuscript. Only a few small details remain in the abstract and in the bibliography that need to be corrected.

Response: We appreciate the positive and valuable comments. In the revised our manuscript, we corrected the errors in the abstract and references. A point-by-point response to each comment is as follows:

3.2 [Comment]: In the abstract it is not necessary to give the confidence intervals of the measurements of the areas or the parameters of statistical significance.

Response: Thanks for this comment. Following your suggestion, we removed the confidence intervals of the measurements of the areas or the parameters of statistical significance.

3.3 [Comment]: The abbreviation MAP is used and has not been previously defined.

Response: We thank the reviewer for this comment .The full name (mean annual precipitation) of MAP was added.

3.4 [Comment]: The final sentence is a speculative statement and can also be deleted.

Response: As suggested, we deleted this sentence from the revised manuscript.

3.5 [Comment]: 3.2.2. In the references list, delete some dots:

Line 386, delete dot in Science. 2005.

Line 404, 426, 466, delete dot in Water. 2015.

Line 451, delete dot in One. 2012.

Line 476, delete dot in Sensors. 2008.

Line 486, delete dot in Systems. 2000.

Line 500, delete dot in Procedia. 2010.

Response: We thank the reviewer for this comment. Done as suggested.